# Genome-wide off-rates reveal how DNA binding dynamics shape transcription factor function

Wim J de Jonge, Mariël Brok, Philip Lijnzaad [ID], Patrick Kemmeren [ID] & Frank CP Holstege[*] [ID]

## Abstract

**Protein–DNA interactions are dynamic, and these dynamics are an important aspect of chromatin-associated processes such as transcription or replication. Due to a lack of methods to study on- and off-rates across entire genomes, protein–DNA interaction dynamics have not been studied extensively. Here, we determine *in vivo* off-rates for the *Saccharomyces cerevisiae* chromatin organizing factor Abf1, at 191 sites simultaneously across the yeast genome. Average Abf1 residence times span a wide range, varying between 4.2 and 33 min. Sites with different off-rates are associated with different functional characteristics. This includes their transcriptional dependency on Abf1, nucleosome positioning and the size of the nucleosome-free region, as well as the ability to roadblock RNA polymerase II for termination. The results show how off-rates contribute to transcription factor function and that DIVORSEQ (Determining *In Vivo* Off-Rates by SEQuencing) is a meaningful way of investigating protein–DNA binding dynamics genome-wide.**

**Keywords** DNA binding dynamics; epigenetics; genomics; systems biology; transcription

**Subject Categories** Chromatin, Transcription & Genomics; Methods & Resources

**Mol Syst Biol. (2020) 16: e9885**

## Introduction

Processes that act on chromatin, such as transcription or replication, are controlled by molecular interactions. This includes proteins interacting with DNA. Protein–DNA interactions are dynamic, and these dynamics are likely important to achieve appropriate regulation of DNA-dependent processes. During transcription for example, different types of transcription factors (TFs) are continuously interacting with chromatin in a variety of ways. Each brings different functions into play: opening or closing chromatin, creating loops, modifying or evicting nucleosomes, recruiting cofactors and, in the case of activation, ultimately causing formation of a pre-initiation complex that includes RNA polymerase (Hahn & Young, 2011; de Laat & Dekker, 2012; Spitz & Furlong, 2012; Struhl & Segal, 2013;

Friedman & Rando, 2015; Kubik *et al*, 2017; Lai & Pugh, 2017; Woo *et al*, 2017; Cramer, 2019; Brahma & Henikoff, 2020). TFs are therefore constantly moving off and onto different loci, probing for appropriate interactions, also under conditions of steady-state transcriptional output (Hammar *et al*, 2014). The rates with which proteins interact with DNA, their on- and off-rates, dictate the outcome of all kinds of regulatory programmes. Understanding how DNA-dependent processes work at the molecular level therefore requires methods to measure the dynamics of protein–DNA binding interactions in a systematic manner.

Different methods have been applied to investigate protein–DNA interaction dynamics. Initial *in vitro* measurements showed very stable TF-DNA binding that could last for more than an hour (Perlmann *et al*, 1990; Hoopes *et al*, 1992). This view was challenged by *in vivo* measurements showing much more dynamic interactions (Hager *et al*, 2009; Larson, 2011; Mueller *et al*, 2013; Voss & Hager, 2014; Coleman *et al*, 2015; Brignall *et al*, 2019; Elf & Barkefors, 2019), likely in part due to the presence of nucleosomes (Luo *et al*, 2014; Donovan *et al*, 2019a; Mivelaz *et al*, 2020). Direct visualization of protein–DNA interaction dynamics by fluorescence microscopy has been pivotal in forming the current view that binding of many proteins is indeed highly dynamic (McNally *et al*, 2000; Elbi *et al*, 2004; Karpova *et al*, 2004, 2008; Stavreva *et al*, 2004; Bosisio *et al*, 2006; Yao *et al*, 2006; Kloster-Landsberg *et al*, 2012). These studies have also been crucial for showing the importance of dynamics and how this can be regulated through distinct mechanisms. A drawback of microscopy is scope however. Information is provided for only part of the nucleus collectively, or only for a single locus. It would be very useful to determine interaction dynamics at many different binding sites individually, preferably across an entire genome.

Genome-wide protein–DNA binding can be measured by chromatin immunoprecipitation (ChIP) *in vivo* (Gilmour & Lis, 1984; Kuo & Allis, 1999; Park, 2009; Collas, 2010; Furey, 2012). On its own, ChIP only provides a static indication of the degree of binding during the time-window of protein–DNA cross-linking. ChIP cannot measure protein–DNA binding dynamics directly. Competition ChIP is a ChIP variant that uses inducible switching between two differentially tagged isoforms of the same protein and has been applied to measure turnover of nucleosomes and TFs (Dion *et al*, 2007; Rufiange *et al*, 2007; van Werven *et al*, 2009; Lickwar *et al*, 2012; Hasegawa & Struhl, 2019). Although limited by the induction

Princess Máxima Center for Pediatric Oncology, Utrecht, The Netherlands
*Corresponding author. Tel: +31 88 972 7272; E-mail: f.c.p.holstege@prinsesmaximacentrum.nl

kinetics of the competing isoform, competition ChIP has nevertheless highlighted the advantage of determining dynamics at different loci in a genome-wide manner. It has revealed differences in dynamics between promoter classes, differences in nucleosome turnover between promoters and gene bodies and showed that differential TF turnover at different loci is an important basis of transcription regulation.

Binding dynamics are determined by TF concentrations, and by on- and off-rates. On- and off-rates are two distinct facets of dynamics. Both would be useful to measure separately since they are likely influenced and regulated by different molecular mechanisms. Having both would also enable the estimation of dissociation constants and the binding free energy. A second adaptation of ChIP has indeed focused on determining on-rates by measuring the kinetics of binding during a formaldehyde cross-linking time–course (Poorey *et al*, 2013). As a method, this is still under development (Zaidi *et al*, 2017) and has only been applied to a few binding sites and not genome-wide as yet. Here, we devised a method that directly determines off-rates and does so for all binding sites across a genome. This was achieved by applying anchor-away to rapidly deplete unbound proteins from the nucleus (Haruki *et al*, 2008; Grimaldi *et al*, 2014), thereby removing the on-rate contribution to binding levels. Monitoring the time-dependent decay of protein–DNA binding across all genomic locations results in determination of off-rates, also in the form of locus-specific mean residence times. The method DIVORSEQ (Determining *In Vivo* Off-Rates by SEQuencing) is applied here to Abf1, a *Saccharomyces cerevisiae* general regulatory factor akin to chromatin pioneering TFs in mammals (Zaret & Carroll, 2011; Kubik *et al*, 2017). Alongside roles in shaping chromatin architecture (Venditti *et al*, 1994; Lascaris *et al*, 2000; Yarragudi *et al*, 2004; Hartley & Madhani, 2009), several different functions have been attributed to Abf1, including involvement in transcription regulation (Gailus-Durner *et al*, 1996), telomere binding (Enomoto *et al*, 1994; Pryde & Louis, 1999), DNA replication (Marahrens & Stillman, 1992), DNA repair (Reed *et al*, 1999) and RNA polymerase II roadblock termination (Roy *et al*, 2016; Candelli *et al*, 2018). Applying DIVORSEQ to Abf1 results in determination of off-rates for 191 different binding sites, with estimated mean residence times ranging from 4.2 to 33 min. Sites with different off-rates are associated with different functional characteristics that include their transcriptional dependency on Abf1, nucleosome positioning and the ability to roadblock RNA polymerase II thereby aiding transcription termination. The results emphasize that off-rate is an important characteristic of TF function and indicate that DIVORSEQ is a useful method for investigating protein–DNA binding dynamics genome-wide.

# Results

### Nuclear depletion of Abf1

To inducibly remove unbound Abf1 from the nucleus, an Abf1 anchor-away strain (Haruki *et al*, 2008) was created in the *Saccharomyces cerevisiae* BY4742 background (de Jonge *et al*, 2017). Abf1 was tagged with an FK506 binding protein–rapamycin binding (FRB) domain for nuclear depletion, green fluorescent protein (GFP) to monitor cellular localization and a V5 epitope for ChIP (Southern

*et al*, 1991). *ABF1* deletion is lethal (Halfter *et al*, 1989; Rhode *et al*, 1989). To investigate whether tagging of Abf1 interferes with its function, growth of tagged strains was compared to the untagged background. Tagging Abf1 has only a slight effect on growth (Fig EV1A), indicating that tagging does not greatly interfere with Abf1 function, as has been observed before (Kubik *et al*, 2015). Because of its essential nature, cells are expected to cease growth when Abf1 is depleted from the nucleus. Indeed, upon inducing nuclear depletion, cells show a clear disruption of growth, leading to complete growth cessation (Fig EV1B). Because loss of growth is a downstream effect, the rate of growth cessation does not necessarily reflect the speed of nuclear depletion (de Jonge *et al*, 2017). To directly visualize depletion, cellular localization of Abf1 was monitored using fluorescence microscopy. As expected, nuclear depletion of Abf1 indeed occurs much more rapidly than growth cessation (Fig EV1C and D).

### Determining *in vivo* off-rates by sequencing: DIVORSEQ

Having ascertained Abf1 depletion, we next determined whether the system can be used to measure TF mean residence times (defined as $1/k_{off}$) at different sites across the genome. As published elsewhere (preprint: de Jonge *et al*, 2019), first the ChIP protocol was extensively optimized at almost all steps, to yield results better comparable between different time points. Next, to determine off-rates, Abf1 was depleted from the nucleus and its binding levels were measured genome-wide using the optimized ChIP-seq protocol at 11 time points during 90 min of depletion, all in biological triplicate (Fig 1A). The binding sites detected before depletion ($t = 0$) correspond well with previously published Abf1 binding sites (Kasinathan *et al*, 2014; Zentner *et al*, 2015; Rossi *et al*, 2018b). Over 90% of binding sites overlap with previously reported sites (Fig EV2A). As exemplified, different genomic locations show distinct rates of binding peak decay (Fig 1B), indicating different mean Abf1 residence times at these sites. Quantification and fitting the exponential decay model (Fig 1A) to the data (Fig 1C–E) yields an estimated site-specific off-rate that can also be expressed as an average TF residence time for that site. The examples (Fig 1B–E) were chosen to cover the wide range of different off-rates/mean residence times observed. First, stringent peak filtering was performed to obtain only reliable signals, by selecting sites with strong binding (fold enrichment > 4) as well as sites that have a G/ C residue at −8 bp from the motif (Fig EV2B) and therefore can efficiently be cross-linked (Rossi *et al*, 2018a). Next, off-rates and the corresponding mean residence times were obtained for these 191 selected Abf1 binding sites by fitting exponential decay models to the ChIP-seq data of each individual binding peak. Almost all models closely match the actual binding data, with low residuals (Fig EV2C) and a median $R^2$ of 0.94 (Fig EV2D, lowest $R^2$ = 0.65). Based on these models, the off-rates for Abf1 range between 0.030 and 0.24 $min^{-1}$ (Fig 1F). This corresponds to a mean residence time of 4.2 min for the most dynamic Abf1 site, the divergent promoter of *SRB2* and *NCP1*, and a mean residence time of 33 min for the promoter of *OCA5* which has a very stable Abf1 binding peak (Fig 1G). This is the first indication that DIVORSEQ can measure mean residence times over a considerable range and that Abf1 has distinct mean residence times at different locations across the genome.

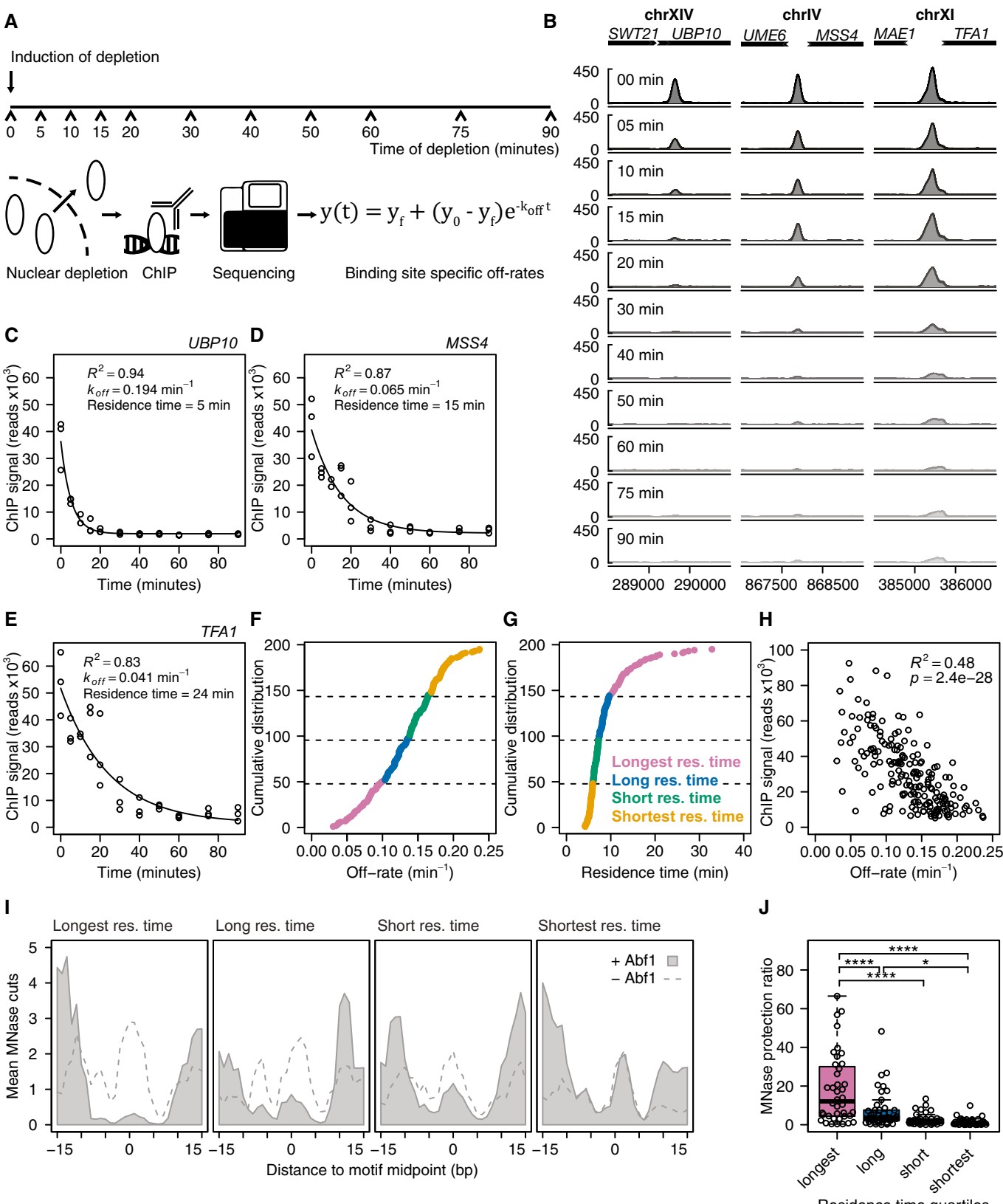

**Figure 1.**

**Figure 1. DIVORSEQ measures distinct mean residence times at different genomic sites.**

A   Schematic overview of the DIVORSEQ method. Unbound protein of interest is depleted from the nucleus, and at several time points during the depletion, binding levels are measured using ChIP-seq. The decrease in binding levels is fitted using an exponential decay model and off-rates and mean residence times are estimated for all binding sites across the genome.

B   Abf1 binding during the depletion time–course at three Abf1 binding sites with different rates of binding decrease. The signal at each time point is the average of three biological replicates, except for the 10 min time point, where one time point was discarded.

C–E  Fit of the exponential decay model for the examples shown in (B). The estimates for off-rates, mean residence time and goodness of fit are shown in the plots.

F, G  Cumulative distribution of the off-rates (F) or mean residence times (G) of the 191 binding sites. The four different mean residence time quartiles are highlighted with different colours.

H   Relationship between Abf1 binding levels before depletion and off-rates for the 191 Abf1 binding sites.

I   Average *in vivo* MNase sensitivity of the mean residence time quartiles, plotted as the average number of MNase cuts at each position relative to the Abf1 binding motif (MacIsaac *et al*, 2006) before (grey fill) or after (dashed line) nuclear Abf1 depletion. The data were smoothed using a 3 bp window.

J   Extent of MNase protection by Abf1 for each mean residence time quartile. The MNase protection ratio is the mean number of cuts in the region protected by Abf1 ($-8$ bp until $+8$ bp) after depletion of Abf1 (I, dashed line), divided by the mean number of cuts in the same region before depletion (I, light grey fill). Asterisks denote a significant difference between the quartiles, calculated using a one-way ANOVA followed by Tukey's honest significant difference (HSD) test (*$P < 0.05$ and ****$P < 0.0001$).

## DIVORSEQ-derived TF mean residence times correspond to MNase protection rates

To initially test whether DIVORSEQ-derived off-rates are realistic measures of Abf1 binding stability, two strategies were employed. First, mean residence times were compared to Abf1 binding at $t = 0$. Since off-rates influence steady-state binding levels, some degree of correspondence is expected. There is indeed correlation between the DIVORSEQ-derived off-rates and binding levels (Fig 1H), whereby sites with low off-rates have higher Abf1 binding levels. That the correspondence is not complete is also expected because on-rate contributes to steady-state binding levels as well. These results therefore also indicate that the relative importance of on- and off-rates may differ for different genomic binding sites. A second verification of the DIVORSEQ-derived off-rates was therefore also sought. This was based on an independently generated MNase cleavage dataset, derived from a strain expressing free MNase (Kubik *et al*, 2018). Since it is well known that TF occupancy can result in protection against MNase cleavage, it is expected that Abf1 binding sites with the lowest off-rate should show the highest degree of MNase protection. This is indeed the case. Abf1 binding sites were divided into the four quartiles with the longest, long, short and shortest mean Abf1 residence times (Fig 1F and G). The average MNase cleavage is plotted for each quartile relative to the Abf1 binding motif (Fig 1I, grey area) and is indeed seen to increase in the four quartiles from left (longest mean residence times, least MNase cleavage) to right (shortest mean residence times, most cleavage). That protection against MNase cleavage in the quartiles with long mean Abf1 residence times is indeed dependent on Abf1 is demonstrated by an overall increase in MNase cleavage after prolonged Abf1 depletion (Fig 1I, dashed line). The extent of MNase protection is also shown for each individual site in each quartile (Fig 1J). DIVORSEQ-derived Abf1 off-rates correspond well to the degree of MNase protection. This indicates that the method performs as designed and provides meaningful data for a wide range of TF binding stabilities at different locations across the genome.

## Increased Abf1 binding stability is associated with larger nucleosome-free regions

Having established that the DIVORSEQ-derived off-rates are meaningful reflections of binding stability, we next asked whether there

are mechanistic relationships between stability as measured in this manner and the roles of Abf1. Abf1 is important for shaping local chromatin architecture (Venditti *et al*, 1994; Lascaris *et al*, 2000; Yarragudi *et al*, 2004; Hartley & Madhani, 2009; Ganapathi *et al*, 2011; Krietenstein *et al*, 2016; Kubik *et al*, 2018) and contributes to the creation of nucleosome-free regions (NFRs) by competing with nucleosomes and acting as a barrier that chromatin remodellers use to position surrounding nucleosomes. To investigate whether Abf1 binding stability is related to its role in creating NFRs, nucleosome positioning data (Kubik *et al*, 2015) were investigated in the context of different mean Abf1 residence times. Sites with more stable Abf1 binding (longer mean residence times, Fig 2, top) have larger NFRs (308 bp) compared to sites with shorter mean residence time sites (Fig 2, bottom, 272 bp). These results obviously fit well with the idea that more stably bound Abf1 can more efficiently repel nucleosomes. However, this does not rule out the converse whereby nucleosome remodelling and nucleosome competition causes increased Abf1 off-rates at those sites with reduced mean residence times. Most importantly for the goals of our study, alongside the MNase protection data (Fig 1I and J), the NFR size associated differences shows that DIVORSEQ-derived Abf1 off-rates can be functionally meaningful in this manner too.

## Changes in mRNA synthesis rates match Abf1 binding dynamics

We next investigated whether Abf1 binding dynamics play a role in the function of Abf1 as a transcriptional regulator (Buchman & Kornberg, 1990; Gailus-Durner *et al*, 1996; Miyake *et al*, 2002, 2004; Yarragudi *et al*, 2007; Paul *et al*, 2015; Kubik *et al*, 2018). Previous studies have shown that not all Abf1 bound promoters show transcriptional dependency on Abf1 (Schroeder & Weil, 1998; Yarragudi *et al*, 2007; Paul *et al*, 2015). This has been ascribed to either lower inherent propensity for nucleosome formation at some sites, binding too far away from a transcription start site, or redundancy with other TFs (Paul *et al*, 2015; Kubik *et al*, 2018). We therefore first determined which genes are dependent on Abf1 by measuring mRNA synthesis rates genome-wide using 4-thiouracil labelling of nascent transcripts (Sun *et al*, 2012) during a 90 min Abf1 depletion time–course, at the same times points that were used to determine off-rates (Fig 3A). Approximately half of the genes with Abf1 promoter binding show a decrease in mRNA synthesis rates upon Abf1 depletion, in agreement with what has previously

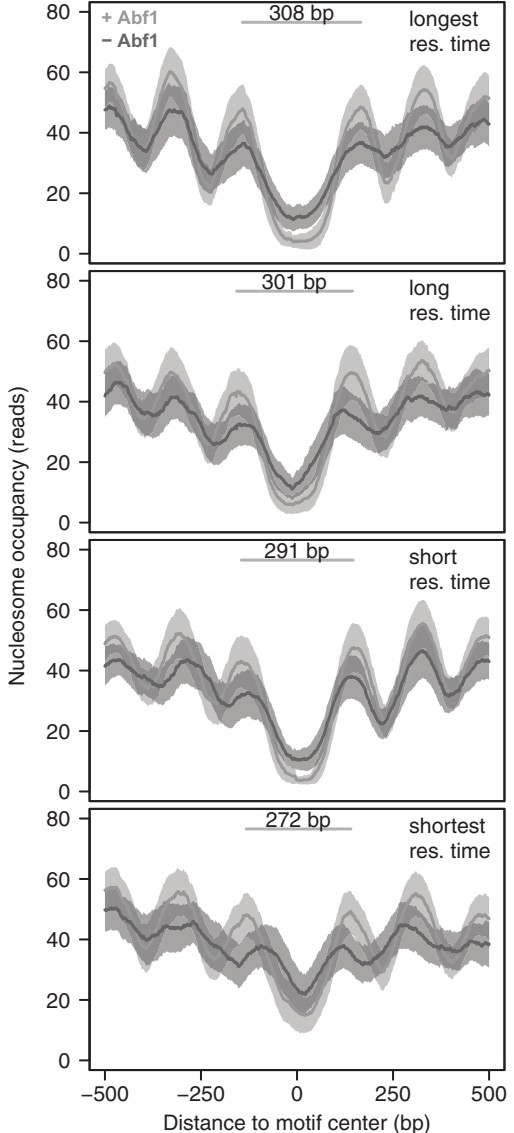

**Figure 2. Nucleosomes architecture corresponds to mean residence times.**

Average nucleosome occupancy of the mean residence time quartiles before (light grey) and after (dark grey) Abf1 depletion, centred on the Abf1 binding motif. Nucleosomes of all mean residence time quartiles reposition upon Abf1 depletion, which indicates that Abf1 contributes to the positioning of nucleosomes for all quartiles. The average distance between the midpoints of the −1 and +1 nucleosomes before depletion of Abf1 are indicated for each quartile. The confidence intervals are indicated by a transparent grey fill, calculated as the mean ± the standard error of the mean (SEM). Nucleosome binding data are from (Kubik *et al*, 2015).

been described for Abf1 (Schroeder & Weil, 1998; Yarragudi *et al*, 2007; Paul *et al*, 2015). The non-responsive genes show little, if any, concomitant change in nucleosome repositioning (Figs 3B and EV3A), also in agreement with previous studies (Kubik *et al*, 2018). The set of genes that do show Abf1-dependency were next used to investigate the role of binding dynamics.

First, steady-state synthesis rates (transcripts per minute per cell, Sun *et al*, 2012) were compared to steady-state binding levels of Abf1. In contrast to what might be expected, there is virtually

no relationship between the amount of Abf1 at a promoter and promoter activity at steady state (Fig 3C). Regardless of absolute binding levels, promoters with more stably bound Abf1 also do not show higher synthesis rates (Fig 3D). There is however some association between the steady-state amount of bound Abf1 and the early changes in synthesis rates observed upon Abf1 depletion (Figs 3E and EV3B). The relationship between Abf1 presence and transcriptional dependency is markedly stronger when taking into account the DIVORSEQ-derived off-rates (Figs 3F and EV3C). Genes showing the largest change in promoter output are those with the highest off-rates. This holds both for the 10 min time point analysed in Fig 3F, as well as when fitting an exponential decay model to the entire mRNA synthesis rate time–course (Fig 3G). For those genes that are dependent on Abf1 for transcriptional activity, there is a strong correspondence between the loss of Abf1 and the reduction in synthesis rate. The relationship between Abf1 and transcriptional output only becomes clear when plotting off-rates (Fig 3F and G). This emphasizes the importance of methods to investigate interaction dynamics genome-wide and the utility of DIVORSEQ for this purpose. As discussed later, our analyses agree with the idea that an Abf1-dependent NFR is the most important determinant for setting up transcription, resulting in associated dependencies on Abf1 (Paul *et al*, 2015; Kubik *et al*, 2018; Fig 3B), but that fine-tuning the absolute levels of steady-state transcriptional output is further dependent on other contributing transcription (co-)factors.

## Stably bound sites are more efficient roadblocks for pervasive Pol II transcription

In addition to being a chromatin organizer, Abf1 has been shown to function as roadblock for pervasive transcription (Roy *et al*, 2016; Candelli *et al*, 2018). In this role, Abf1, like the other general regulatory factors Reb1 and Rap1, can block transcribing RNA polymerase II (Pol II). This collision causes Pol II to stall, to become ubiquitinylated and likely degraded (Colin *et al*, 2014; Candelli *et al*, 2018). An obvious hypothesis, as has indeed been suggested (Roy & Chanfreau, 2018), is that TF binding stability may contribute to roadblock function. To test this idea, data of actively transcribing Pol II (Schaughency *et al*, 2014) were analysed. The average Pol II presence relative to the Abf1 binding motif was plotted for each of the mean residence time quartiles (Fig 4A). A roadblock peak (Fig 4A, arrow) can be observed immediately upstream of the Abf1 motif for the quartile with the longest mean residence times, and this becomes less pronounced with shorter mean residence times. Quantification of roadblock efficiency at each individual site confirms that stronger roadblocks are observed in the quartile with the longest mean residence times (Fig 4B). In agreement with this, sites with stalled Pol II have significantly lower off-rates compared to sites that do not (Fig 4C). More stably bound Abf1 is a more efficient roadblock for transcribing Pol II, further demonstrating the utility of genome-wide off-rate measurements for molecular mechanistic understanding.

## Factors contributing to Abf1 binding stability

When applied to Abf1, DIVORSEQ indicates that there is a considerable range of off-rates and that this contributes to different aspects

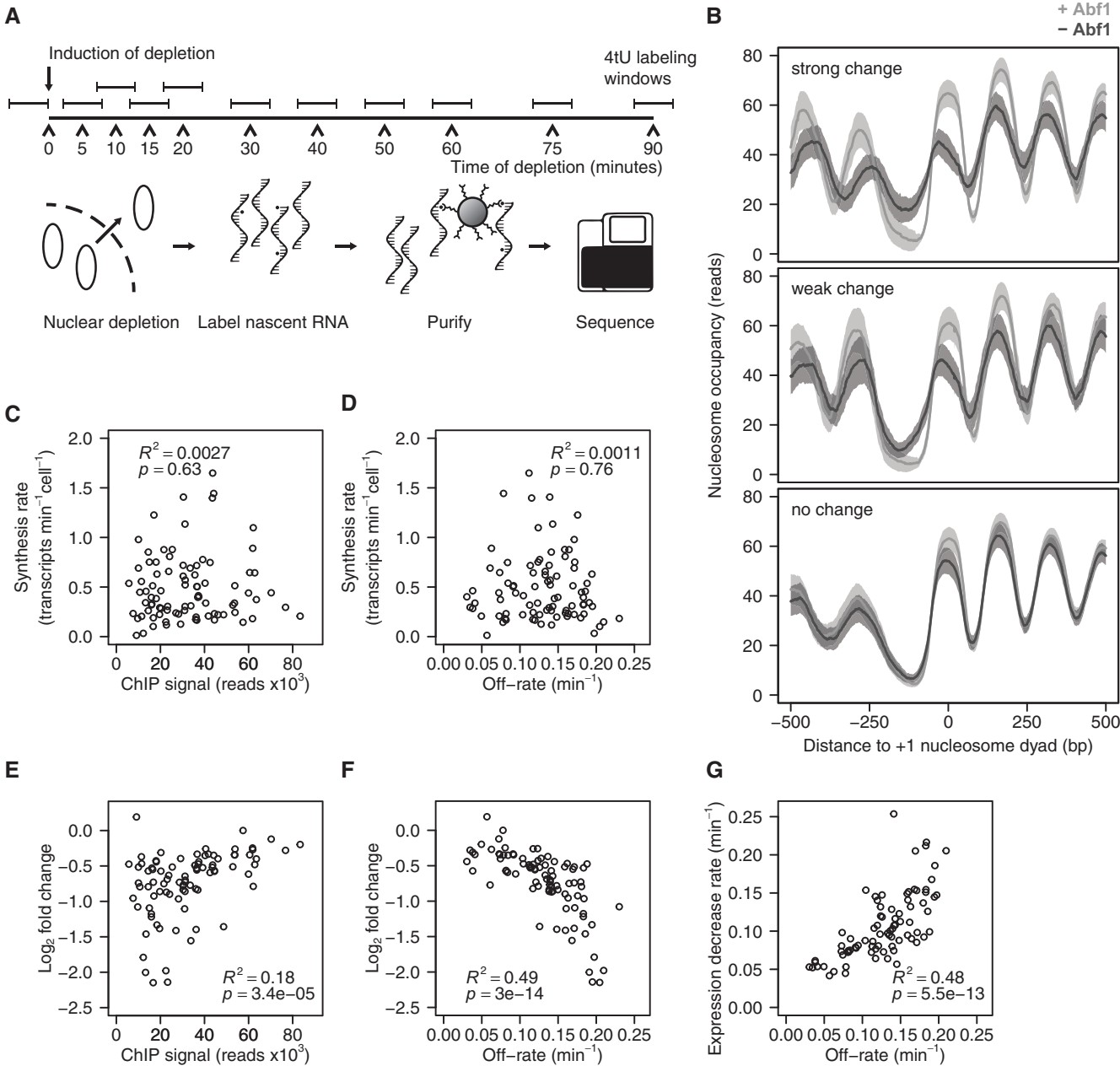

**Figure 3. Dynamics of mRNA synthesis changes follow the dynamics of Abf1 dissociation.**

A   Schematic overview of the experiment set-up for measuring promoter output dynamics by labelling nascent RNA. The protein of interest is depleted from the nucleus and nascent RNA is labelled for 6 min using 4-thiouracil (4tU) at several time points during the depletion. Total RNA is extracted, nascent RNA is purified by biotinylating 4tU labelled RNA and the purified RNA is sequenced. Samples were taken such that the centre of the labelling period was the same as the time points that were used for DIVORSEQ.

B   Average nucleosome occupancy relative to +1 nucleosome dyad, of genes with Abf1 binding to the promoter that show strong changes (fold change > 2 at $t$ = 20, top panel, $n$=44), weak changes (1.5 ≤ fold change ≤ 2 at $t$ = 20, middle panel, $n$ = 42) or no changes (fold change < 1.5 at $t$ = 20, bottom panel, $n$ = 112) in mRNA synthesis upon Abf1 depletion. Nucleosome occupancy is shown before (light grey) and after (dark grey) Abf1 depletion. The confidence intervals are indicated as in Fig 2 by a transparent grey fill, calculated as the mean ± SEM. Nucleosome binding data and +1 nucleosome positions are from (Kubik et al, 2015). Downregulated and Abf1 bound genes without an annotated +1 nucleosome were omitted from the plots.

C, D   Relationship between steady-state synthesis rates (Sun et al, 2012) and binding levels before depletion (C), or off-rates (D). Genes are shown that are Abf1 bound and downregulated (fold change > 1.5 and $P$ < 0.01 at 20 and 30 min of depletion, yielding 88 genes) and have available synthesis rates ($n$ = 87).

E, F   Relationship between $\log_2$ mRNA synthesis rate changes after 10 min of depletion and binding levels before depletion (E) or off-rates (F). Genes are shown that are Abf1 bound and downregulated (fold change > 1.5 and $P$ < 0.01 at 20 and 30 min of depletion, $n$ = 88).

G   Relationship between expression decrease rates of downregulated genes and off-rates of the corresponding Abf1 binding site. The expression decrease rates were calculated by fitting the 4tU-seq time data course using the same exponential decay model that was used for the off-rates. The genes from (E-F) are shown, except for the ones where the 4tU-seq data could not be fitted with an exponential decay model ($n$ = 82).

   

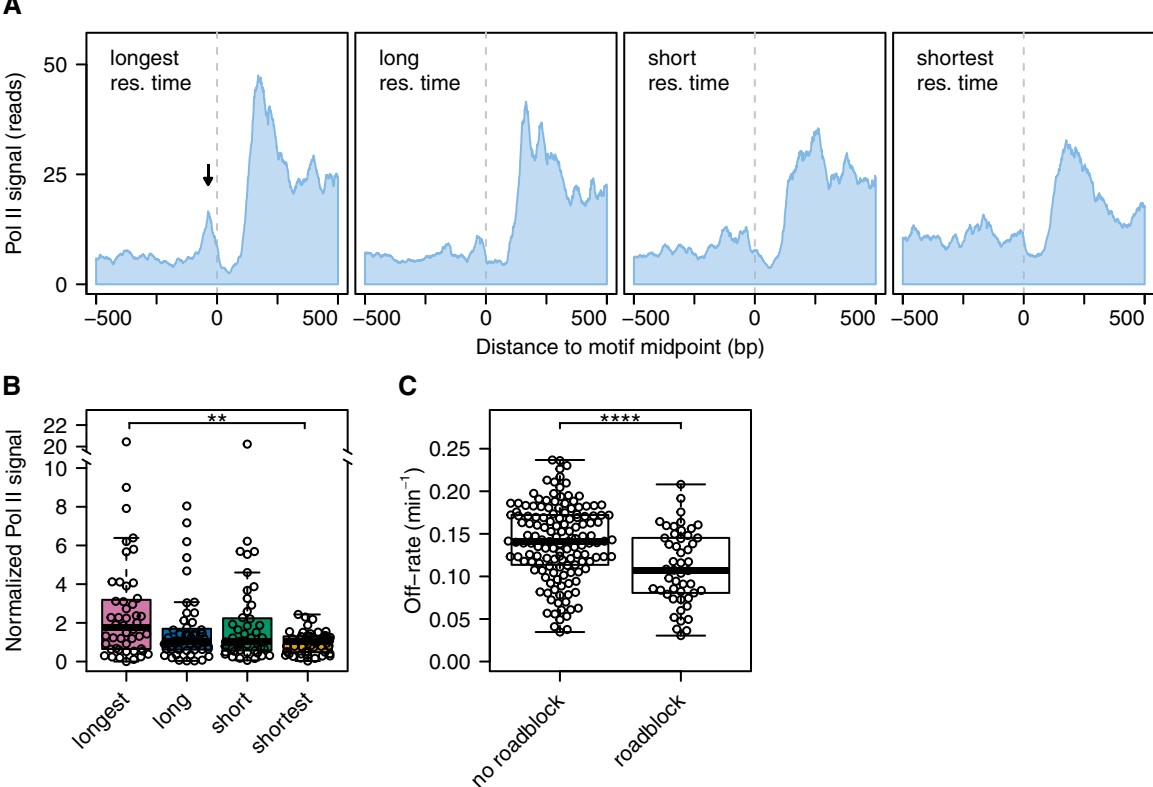

**Figure 4. Sites with long mean residence times function as a roadblock for pervasive transcription.**

A  Average RNA polymerase II binding for the different mean residence time quartiles. Binding profiles are centred on the Abf1 motif. The dotted line marks the midpoint of the Abf1 motif. All RNA polymerase II binding data were reoriented such that each motif is oriented in the same direction. RNA polymerase II binding is PAR-CLIP (photoactivatable ribonucleoside-enhanced cross-linking and immunoprecipitation) data from (Schaughency et al, 2014).

B  Quantification, by mean residence time quartile, of normalized Pol II binding levels at the roadblock peak. These levels are defined as the amount of roadblocked Pol II (located at $-37 \pm 5$ bp) divided by the amount of upstream Pol II (from $-300$ to $-100$ bp). Asterisks denote a significant difference between the quartiles, calculated using a one-way ANOVA followed by Tukey's HSD test ($**P < 0.01$).

C  Difference in off-rates between sites that are a roadblock (normalized Pol II signal > 2) and those that are not. The $P$-value was calculated using a two-tailed $t$-test ($****P < 0.0001$).

---

of Abf1 function. The ability to determine mean residence times also allows for investigation into the factors that determine different off-rates. The DNA binding motif is obviously an important factor for determining Abf1 binding stability. To evaluate the contribution of motif frequency, the number of Abf1 motifs in the vicinity of each Abf1 peak was counted. Although there are only a few peaks with multiple motifs, it is clear that most sites with more than one Abf1 binding motif have significantly longer mean residence times compared to sites with only a single binding motif (Fig 5A). Such increases in stability are likely caused by different types of cooperative effects associated with the presence of multiple motifs (Adams & Workman, 1995; Polach & Widom, 1996; Miller & Widom, 2003; Hager et al, 2009; Mirny, 2010).

Beside the number of motifs, the sequence composition of the binding motif is also likely to contribute to binding stability. To investigate how motif composition affects Abf1 binding stability, the motif score of each of the binding motifs was compared between the mean residence time groups. Binding sites with the longest mean residence time have a motif that is closer to the consensus compared to the other sites (Fig 5B), which indicates that having a stronger

binding motif leads to more stable binding. Indeed, mutating the binding site of another well-studied TF, Gal4, results in a shorter residence time (Donovan et al, 2019b). To investigate the contribution of motif sequence to Abf1 binding stability in more detail, the consensus motifs of the different mean residence time groups were compared to each other (Fig 5C). Although the consensus motifs of all four groups are similar, showing good correspondence to published motifs (MacIsaac et al, 2006; Kasinathan et al, 2014; Zentner et al, 2015; Rossi et al, 2018a), the longest mean residence time group has a significant enrichment ($P = 0.0046$) for a thymine in the variable part of the motif, at position $-1$ bp (Fig 5C, arrow). This suggests that having a thymine at this position is not needed for binding per se, but that it contributes to the binding stability of Abf1. In agreement with this, mutation of a thymine residue at this position reduces the binding levels of Abf1 *in vitro* (Gailus-Durner et al, 1996).

DNA shape can also influence Abf1 binding levels (Zentner et al, 2015; Rossi et al, 2018a). Strongly and weakly bound Abf1 sites differ in their predicted minor groove width as estimated across naked DNA motifs (Rossi et al, 2018a). The Abf1 sites

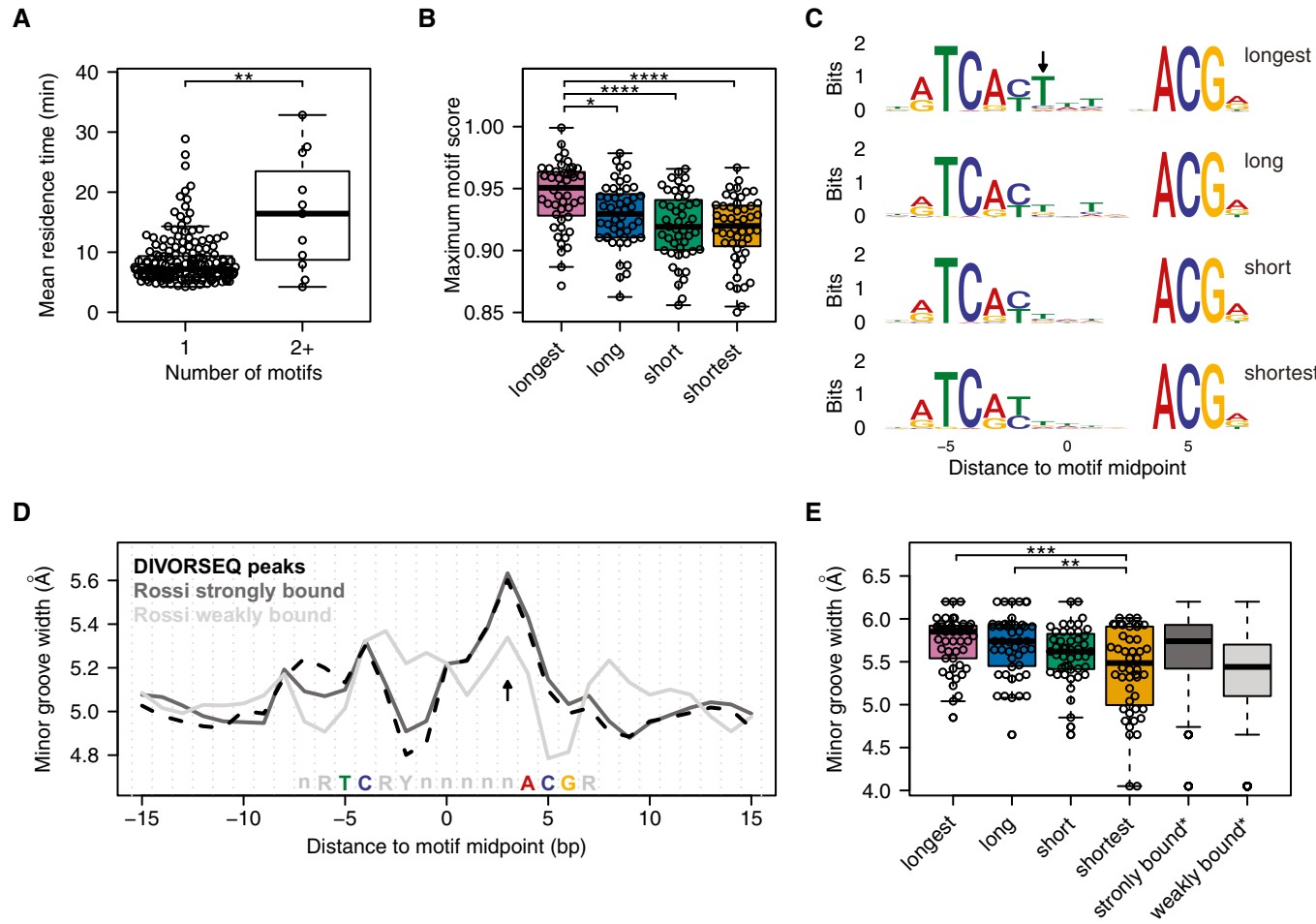

**Figure 5. Factors that contribute to Abf1 binding stability.**

A   Difference in mean residence times between sites with one motif and sites with two or more motifs. The *P*-value was calculated using a Wilcoxon rank-sum test
    (**P = 0.0066), rather than a *t*-test (P = 0.022) as used in Fig 4C, since the group with 1 binding motif is not normally distributed.
B   Difference in maximum motif score between the different mean residence time quartiles. When a site has more than one motif, the highest score was used.
C   Sequence logos showing the representative binding motif of each mean residence time quartile. Motifs from the longest mean residence time quartile are enriched
    (P = 0.0046) for having a thymine at position −1 bp (arrow).
D   Predicted minor groove width centred at the Abf1 binding motif of all 191 Abf1 sites found here (black, dashed line) and Abf1 binding motifs defined as strongly
    bound (dark grey line) and weakly bound (light grey line) by (Rossi *et al*, 2018a).
E   Difference in predicted minor groove width between the mean residence time quartiles at position +3 bp from the motif midpoint (D, arrow). In addition, the minor
    groove width at this position of the Abf1 motifs defined as strongly and weakly bound by (Rossi *et al*, 2018a) are shown (*n* = 400 for each group). Asterisks in (B) and
    (E) denote adjusted *P*-values calculated using a one-way ANOVA followed by Tukey's HSD test between the mean residence time quartiles (*P < 0.05, **P < 0.01,
    ***P < 0.001, ****P < 0.0001).

found here closely resemble the strongly bound sites in their minor groove width pattern (Fig 5D). Strikingly, the same analysis performed across the four groups of sites with different mean residence times reveals that in terms of minor groove width at position +3 bp (Fig 5D, arrow), the group with shortest mean Abf1 residence times most closely resembles the sites with low Abf1 binding levels (Rossi *et al*, 2018a), in having a smaller minor groove (Fig 5E). This suggests that the lower binding levels observed on sites with a reduced minor groove width at the +3 position is caused by a higher off-rate. The extended genome-wide survey of Abf1 binding stabilities demonstrates that factors influencing TF binding stability *in vivo* can also be advantageously studied by DIVORSEQ.

## Discussion

The dynamics of proteins interacting with DNA are thought to play an important role in the regulation of chromatin-associated processes such as transcription (Hager *et al*, 2009). To fully understand these processes at a molecular level requires an understanding of the underlying binding dynamics. DIVORSEQ quantifies protein–DNA binding dynamics *in vivo* by directly measuring off-rates and mean residence times at multiple binding sites across an entire genome. Applying DIVORSEQ to the TF Abf1 shows that the method can measure meaningful differences in off-rates spanning a wide range of values. Our results show how motif number, sequence and structure of the binding motif contribute to off-rates

and how this aspect of binding dynamics influences the roles of Abf1 as a chromatin organizing factor, a transcriptional regulator and a termination roadblock for RNA polymerase II at different sites across the genome.

Abf1 is a general regulatory factor, known to organize chromatin (Venditti *et al*, 1994; Lascaris *et al*, 2000; Yarragudi *et al*, 2004; Hartley & Madhani, 2009; Ganapathi *et al*, 2011; Krietenstein *et al*, 2016; Kubik *et al*, 2018). Our results show good correspondence between NFR size and binding stability (Fig 2). Sites with shorter mean Abf1 residence times have smaller NFRs. This could either be explained by stronger nucleosome exclusion through more stably bound Abf1, or conversely, by more stable binding of Abf1 in larger NFRs. Abf1 shapes the local chromatin architecture by competing with nucleosomes (Venditti *et al*, 1994; Yarragudi *et al*, 2004) and by acting as a barrier that chromatin remodellers use to position flanking nucleosomes (Krietenstein *et al*, 2016). It seems reasonable that Abf1 forms a more efficient barrier when it is more stably bound, thus repelling nucleosomes more efficiently, which would support the hypothesis that stable binding creates bigger NFRs. On the other hand, being a barrier means that chromatin remodellers actively position nucleosomes towards Abf1. Therefore, nucleosomes that are being positioned by remodellers may exert a force on Abf1 and destabilize its binding. In this hypothesis, competition with nucleosomes could reduce the residence time of Abf1, as has been shown for Rap1 (Lickwar *et al*, 2012; Mivelaz *et al*, 2020). Upon depletion of Abf1, nucleosomes become repositioned in all mean Abf1 residence time quartiles, but sites with the shortest mean residence time show the biggest reduction in NFR size (Fig 2). This fits with nucleosome positioning leading to shorter mean Abf1 residence times at these sites. Neither hypothesis can, nor need be excluded as yet. The observed correspondence between NFR size and Abf1 off-rates at different sites highlights the advantage of such measurements as a starting point for detailed characterization of molecular mechanisms.

Besides organizing chromatin, Abf1 also functions as a transcriptional regulator (Buchman & Kornberg, 1990; Gailus-Durner *et al*, 1996; Miyake *et al*, 2002, 2004; Yarragudi *et al*, 2007). Abf1 is known to be only a weak activator of transcription (Buchman & Kornberg, 1990; Levo *et al*, 2017), and the results presented here indicate that it mainly regulates transcription through repositioning nucleosomes, which is consistent with previous reports (Paul *et al*, 2015; Kubik *et al*, 2018). Our results fit with Abf1 stimulating transcription by creating an NFR that allows other regulatory factors to bind and whose activities may more directly dictate steady-state expression levels. This offers an explanation for why there is little correlation between steady-state mRNA synthesis rates and Abf1 binding levels or off-rates (Fig 3C and 3D), but nevertheless good correlation between off-rates and changes in mRNA synthesis rates upon depletion (Fig 3F). Removal of Abf1 causes NFR collapse for those promoters that have no redundant mechanisms of NFR upkeep (Fig 3B), resulting in cessation of promoter activity. This is in contrast to Rap1, which is known to directly contact TFIID (Garbett *et al*, 2007) and may directly recruit the transcription pre-initiation complex itself. Such direct recruitment suggests that Rap1 is the main regulator of transcription of its targets, explaining correlation between Rap1 binding dynamics and steady-state mRNA synthesis levels (Lickwar *et al*, 2012). As with the analysis of roadblock function for Abf1, different modes of regulator

activity or function may therefore be revealed by detailed analyses of binding dynamics.

A limitation of DIVORSEQ is that analysis is in bulk, rather than at the single cell resolution available through microscopy (Hager *et al*, 2009; Larson, 2011; Mueller *et al*, 2013; Voss & Hager, 2014; Coleman *et al*, 2015; Brignall *et al*, 2019; Elf & Barkefors, 2019). This is offset by the advantage of determining off-rates for many loci across the genome in parallel. Other methods that measure site-specific *in vivo* binding dynamics include competition ChIP, which determines turnover and is limited by a slow induction of the competitor protein, as well as by the substantial carbon source perturbation required for induction (Schermer *et al*, 2005). DIVORSEQ directly measures off-rates and has been designed for application alongside on-rate measurement by cross-linking kinetic analyses (Poorey *et al*, 2013), that has yet to be applied at the genomic scale (Zaidi *et al*, 2017). In theory, on-rates can also be inferred from a combination of binding levels and off-rates. Such inference will lead to large estimation errors and therefore direct measurement of on-rates, for example by an approach like cross-linking kinetic analysis is preferable.

Considerations that need to be made when applying DIVORSEQ include having sufficiently rapid removal of unbound proteins from the nucleus. Using anchor-away (Haruki *et al*, 2008), an estimated 2,000 molecules can be depleted from the nucleus per minute (Warner, 1999). This implies that for highly abundant proteins such as Abf1, with an estimated 6,000 molecules (Ho *et al*, 2018), mean residence times will be determined at minute-scale resolution. In other words the lower limit of detecting residence times is mainly determined by the number of TF molecules and the 4 min shortest residence time for Abf1 is the limit for applying DIVORSEQ in its current form to Abf1. The lower limit of residence times determined will be lower for less abundant proteins.

The timescale of minutes found here for Abf1 by DIVORSEQ may seem in contrast to the timescale of seconds that have frequently been reported for other TFs using microscopy based methods (McNally *et al*, 2000; Bosisio *et al*, 2006; Karpova *et al*, 2008; Loffreda *et al*, 2017; Donovan *et al*, 2019a). The microscopy based methods certainly also have their limitations such as sensitivity to the model used for fitting, bleaching and out-of-focus movement, making it challenging to accurately measure minute time scale residence times. Nevertheless, it is important to point out that minute-scale residence times have been reported for several TFs previously (Yao *et al*, 2006; Hammar *et al*, 2014; Agarwal *et al*, 2017; Hansen *et al*, 2017; Donovan *et al*, 2019a; Mivelaz *et al*, 2020). It is possible that different approaches are suitable for different TFs with different ranges of residence times. This issue can best be addressed by comparing measurements of many different TFs by several different methods. Until such a comparison is carried out it is possible that some methods, including DIVORSEQ, should only be interpreted on a relative scale. Such relative measurements are of course still valuable for investigating biological meaningful differences in residence times.

That anchor-away is sufficiently rapid for Abf1, is indicated by the excellent fit to first-order kinetics observed and the wide range of different off-rates obtained. Besides anchor-away, other techniques that facilitate nuclear depletion could also be used (Klemm *et al*, 1997; Bayle *et al*, 2006; Busch *et al*, 2009), contingent on rapidity. A second consideration is that the ChIP or genomic

location protocol requires results that are comparable between time points. Here we first extensively optimized almost all ChIP protocol steps to achieve this (preprint: de Jonge *et al*, 2019). Some limitations still remain. Abf1 can only be cross-linked to sites with a guanine or cytosine at −8 bp from the binding motif centre (Rossi *et al*, 2018a). Sites without a guanidine or cytosine at this position were therefore excluded here although some nevertheless yielded low signals. These signals at sites without a G/C at position −8 bp from the motif did not deplete over time and are likely caused by small amounts of cytosolic Abf1 rebinding during the ChIP procedure. Therefore no reliable fits can be obtained for these sites and they were therefore excluded from all analyses. Combined with peak filtering for robust binding, this reduced the number of Abf1 sites for which off-rate could be determined. Improvements to DIVORSEQ could therefore be aimed at preventing rebinding and/or applying assays that do not depend on cross-linking (Zentner *et al*, 2015; Skene & Henikoff, 2017). These considerations aside, that the determined off-rates are accurate is corroborated by the MNase protection levels at Abf1 sites with different off-rates, as well as by the diverse aspects of previously established Abf1 function presented here for the first time in the context of a large number of genomic binding sites and their binding dynamics.

# Materials and Methods

### Strains

The strains used in this study are *Saccharomyces cerevisiae* anchor-away strains (Haruki *et al*, 2008) that were recreated in the BY4742 background (de Jonge *et al*, 2017). Besides the FRB domain from mammalian target of rapamycin (mTOR), the anchor-away tag consists of yeast enhanced green fluorescent protein (yEFGP) and a triple V5 tag. The parental BY4742 anchor-away strain, which has an *FPR1* deletion and a *tor1-1* mutation to desensitize the strain to rapamycin, was used as a wildtype control.

### Growth conditions

Strains were streaked from −80°C stocks on appropriate selection plates (for the parental anchor-away strain: YPD + Nourseothricin and for the Abf1-aa strains: YPD + Nourseothricin + Hygromycin), and incubated at 30°C for 3 days. In the morning, liquid pre-cultures were inoculated in 1.5 ml of synthetic complete (SC) medium: 2 g/l dropout mix complete and 6.71 g/l yeast nitrogen base without amino acids, carbohydrate & w/AS (YNB) from US Biologicals (Swampscott, USA) with 2% D-glucose. In the afternoon, several pre-cultures were combined, diluted to final volume of 20 ml and grown overnight. The growth conditions were identical for all experiments and pre-cultures: in SC medium at 30°C, with shaking (230 rpm).

### Anchor-away depletion

At $t = 0$, Abf1 was depleted from the nucleus by addition of rapamycin (LC Laboratories #R-5000; dissolved to 2mM in DMSO), to a final concentration of 7.5 μM. For the $t = 0$ time point, the same volume of DMSO instead of rapamycin was added and incubated for 90 min.

### Chromatin immunoprecipitation

ChIP was performed as described in detail in (de Jonge *et al*, 2020) using biological triplicates. To summarize: cells were diluted in the morning to an optical density (OD) of 0.11–0.15 (WPA Biowave CO8000 Cell Density Meter) in 100 ml of SC medium, and grown for at least 2 doublings to an $OD_{600} = 0.8$, which corresponds to about $2 \times 10^7$ cells per ml. Additions of rapamycin and DMSO were staggered such that all time points were ready at the same OD (0.8). When this OD was reached, the cells were cross-linked for 5 min by addition of 37% formaldehyde (Sigma-Aldrich #252549) to a final concentration of 2%. The formaldehyde was quenched using a final concentration of 1.5M of Tris (tris(hydroxymethyl)aminomethane) for 1 min. Subsequently, the cells were pelleted by centrifugation at 3,220 $g$ at 4°C for 3 min. The pellet was washed in 10 ml TBS (150 mM NaCl, 10 mM Tris pH 7.5) and pelleted again at 3,220 $g$ for 3 min at 4°C. After resuspension in 1 ml MQ, cells were centrifuged at 3,381 $g$ for 20 s at room temperature and the pellet was snap-frozen in liquid nitrogen and stored at −80°C.

To lyse cells, the cells were resuspended in FA lysis buffer (50 mM HEPES-KOH pH 7.5, 150 mM NaCl, 1 mM EDTA pH 8.0, 1% Triton X-100, 0.1% Na-deoxycholate, 0.1% SDS) containing the protease inhibitors aprotinin, pepstatin A, leupeptin and PMSF to a final volume of 2ml, transferred to 2-ml screw-cap tubes and disrupted using zirconium/silica beads 0.5 mm (BioSpec Products, #11079105z) by bead beating 7 times 3 min in a Genie Disruptor (Scientific Industries). The lysate was recovered and centrifuged at 1,503 $g$ for 2 min at 4°C to remove cell debris. The supernatant was subsequently fragmented by sonicating the samples for 10 cycles of 15 s on, 30 s off using a Bioruptor Pico sonicator (Diagenode #B01060010).

For the immunoprecipitation, 450 μl of the fragmented chromatin was incubated with 1 μl of anti-V5 antibody (Life Technologies #R96025) for 2 h at 4°C. A 20 μl aliquot was kept separate as an input control. The chromatin + antibody were subsequently bound for 20 min at room temperature to magnetic beads (Dynabeads protein G, Life Technologies #10004D) that were pre-incubated with BSA. The beads were washed twice with PBS and twice using PBS-T. During the last wash, the beads were transferred to fresh LoBind tubes (Eppendorf #0030108051). Cross-links were reversed by incubating in TE/1% SDS (10 mM Tris pH 8.0, 1 mM EDTA pH 8.0, 1% SDS (w/v)) overnight at 65°C. The next morning, RNA was degraded by addition of RNAse A/T1 (Thermo Scientific #EN0551) at 37°C, and subsequently proteins were digested by addition of proteinase K (Roche #03115852001) at 37°C. After protein digestion, DNA was recovered using a Qiagen PCR purification cleanup kit (Qiagen #28106) by eluting in 30 μl buffer EB.

### RNA labelling and extraction

For the 4tU-seq time–course, 20 ml cultures were used, with biological triplicates for each time point. Rapamycin and DMSO additions were staggered such that all cultures were ready at the same OD (0.8). WT samples incubated with rapamycin or DMSO for 90 min were taken along as a controls. Three minutes before the cultures were ready, 4-thiouracil (4tU; Sigma-Aldrich #440736) was added to the cell cultures to a final concentration of 5 mM. Cells were incubated with 4tU for 6 min in total, such that the centre of the

labelling period matched the time point of the ChIP time–course. Subsequently, cells were harvested by centrifugation at 3,220 $g$ for 3 min, cell pellets were snap-frozen immediately in liquid nitrogen and stored at −80°C.

To isolate total RNA, cells were resuspended in Acid Phenol Chloroform (Sigma #P1944) and immediately mixed with the same volume of TES buffer (10 mM Tris pH 7.5, 10 mM EDTA, 0.5% SDS). The samples were vortexed hard for 20 s, the tubes were covered in aluminium foil to keep the samples dark, and incubated in a 65°C water bath for 10 min. Next, the samples were transferred to 1.5 ml tubes and incubated in a thermomixer for 50 min at 65°C and 1,400 rpm, while covered with aluminium foil. After incubation, samples were centrifuged at 18,407 $g$ for 20 min at 4°C. The water phase was recovered and phenol extraction was repeated once, followed by extraction using chloroform–isoamyl alcohol (25:1). The RNA was precipitated using sodium acetate (NaAc 3M, pH 5.2) and 100% ethanol (−20°C) by incubating at −20°C for at least 30 min. DTT was also added to a final concentration of 1 mM to prevent oxidation of the 4tU. The pellet was washed once with 80% ethanol and resuspended to a final concentration of 1 μg/μl in sterile MQ.

To recover nascent transcripts the protocol from (Dölken *et al*, 2008) was used with minor adaptations. In brief, 100 μg of cleaned RNA was heated to 60°C for 10 min and immediately put on ice for 2 min. The RNA was biotinylated by adding 200 μl Biotin-HPDP (Thermo Fisher Scientific #21341) dissolved to 1 mg/ml in 30% DMF. Unbound biotin was removed using chloroform extraction. Biotinylated RNA was separated from total RNA using streptavidin-conjugated magnetic beads and μMACs columns (Miltenyi Biotec #130-074-101). The beads were washed 6x using 65°C washing buffer (100 mM Tris pH 7.5, 10 mM EDTA, 1 M NaCl, 0.1% Tween-20) and bound RNA was eluted using 200 μl of 100 mM DTT. The nascent RNA was purified using an RNeasy MinElute Cleanup Kit (Qiagen #74204).

### Library preparation and sequencing

ChIP-seq libraries were created using a combination of a NEXTflex Rapid DNA-Seq Kit (Bioo Scientific #NOVA-5144) and a NEXTflex Rapid Directional qRNA-Seq Kit (Bioo Scientific #NOVA-5130) to allow for incorporation of unique molecular identifiers (UMIs) (Kivioja *et al*, 2012) on both sides of each fragment. To improve speed and accuracy, a maximum of 8 libraries were prepared at the same time. To make the amount of starting material of the input samples similar to that of the IP samples, the input samples were diluted 1:300 prior to library prep. End-repair and adenylation were carried using the NEXTflex Rapid DNA-Seq Kit with half of the recommended volumes. The subsequent steps were carried out using the NEXTflex Rapid Directional qRNA-Seq Kit with a quarter of the recommended volumes for the adapter ligation and half volumes for the PCR amplification. The initial volume used for each of the bead clean-ups was adjusted to 50 μl by addition of MQ and bead ratios were kept as recommended. The number of PCR cycles used was the same for all ChIP and input samples (13 cycles) except for the WT IPs where 15 PCR cycles were used. Library yields were assessed using a High Sensitivity DNA bioanalyzer chip (Agilent) and equimolar amounts of library were pooled and sequenced paired-end 2 × 75 bp on a NextSeq 500 system (Illumina).

4tU-seq libraries were created using the NEXTflex Rapid Directional qRNA-Seq Kit (Bioo Scientific #NOVA-5130) with a slightly modified protocol. During step *D* of the protocol (*i.e.* bead cleanup after second strand synthesis) the beads were resuspended in 10 μl of resuspension buffer, and 8 μl was used for the next step. From this step onwards, half of all recommended volumes were used. The initial volume used for each of the bead clean-ups was adjusted to 50 μl by addition of MQ and bead ratios were kept as recommended. Since the concentrations of labelled RNA differed after the purification, a qPCR was performed to estimate the number of PCR cycles needed for each sample after adapter ligation. The number of cycles that were used varied between 8 and 12 cycles. Library yields were assessed using a High Sensitivity DNA bioanalyzer chip (Agilent) and equimolar amounts of library were pooled and sequenced paired-end 2 × 75 bp in two sequence runs on a NextSeq 500 system (Illumina).

### Mapping

Reads from both the ChIP-seq and 4tU-seq experiments were aligned to the sacCer3 genome assembly (February 2011) using HISAT2 v2.0.5 (Kim *et al*, 2015). The settings for the ChIP-seq samples were `"--add-chrname -X 1000 --score-min L,0, -0.2 -k 1 --no-spliced-alignment -5 12 -3 6"` and for the 4tU-seq samples the settings `"--add-chrname -X 1000 --score-min L,0,-0.175 -5 10 -3 10 --dta -max-intronlen 1500 --rna-strand-ness RF"` were used. Subsequently, the bam files were filtered to keep only transcripts with a unique combination of UMIs using the custom scripts "addumis2bam.sh" and "uniqify-umis.pl" available from https://github.com/wdejonge/DIVORSEQ.

### Peak calling and filtering

The ChIP-seq data were filtered to keep only uniquely mapping reads and subsequently peaks were called using MACS2 v2.1.1.20160309 (Zhang *et al*, 2008) with the settings `"-f BAMPE -g 1.25e7 --keep-dup all --mfold 5 2000 --call-summits -q 0.001 --fe-cutoff 2"` using all three replicate $t = 0$ time points (no depletion) versus their corresponding inputs, yielding 948 Abf1 peaks.

To monitor the depletion, the initial binding levels need to be sufficiently strong to accurately measure a reduction in binding levels. Therefore, only binding peaks with a fold enrichment of at least 4 were considered (421 peaks). This also filters out apparent weak binding across open reading frames and tRNAs, which is a known artefact of ChIP (Park *et al*, 2013; Teytelman *et al*, 2013). In addition, it has been recently shown that Abf1 is only efficiently cross-linked to sites with either a guanine or a cytosine at −8 bp from the centre of the Abf1 binding motif (Rossi *et al*, 2018a). For each binding peak, we searched for motifs (±100 bp from peak summit) that closely match the Abf1 consensus (at least 85% of the consensus motif score from (MacIsaac *et al*, 2006)) and determined whether there was a G/C or an A/T at −8 bp from each motif. Only peaks where all motifs found had a G/C at −8 bp from the motif midpoint were kept for further analysis (195 peaks). Four of these peaks were located in telomeric regions. Although the mean residence time estimates of these four peaks are probably accurate, we

noticed that other characteristics were very distinct from other peaks (*e.g.* nucleosome organization, motif composition, RNA polymerase II binding). Therefore, these peaks were excluded, yielding a total of 191 Abf1-binding sites that were analysed in greater detail.

## Exponential decay fitting

To fit the exponential decay model, first the bam files were centred and smoothed with a 101 bp window, as described in (de Jonge *et al*, 2017). Subsequently, all samples were scaled to 1 million reads using genomecov from the bedtools2 suite v2.27 (Quinlan & Hall, 2010), which makes sure that total coverage across the genome is the same for all samples ($1.01 \times 10^8$ bases). For each binding site ($n = 191$), the total coverage was calculated for each sample in a window $\pm$ 50 bp from the peak summit. Subsequently, the binding at all peaks was normalized to the background levels. This was achieved by first calculating the fraction of background reads of each sample, by dividing the number of reads not present in any of the 948 peaks by the total number of reads. The binding levels at each peak were then divided by this fraction of background reads, essentially equalizing the background level of each sample. These values were used to fit a first-order exponential decay function using the `nls` function in R:

$$y(t) = y_f + (y_0 - y_f)e^{-k_{off}t} \qquad (1)$$

with $y_0$ the binding levels a $t = 0$ (*i.e.* before depletion), $t$ the time since depletion, $y_f$ the final binding level and $k_{off}$ the decay rate of the binding levels. The use of a site-specific $y_f$ is needed since different sites decay to a different background levels. The mean residence time is given by $1/k_{off}$ and represents the average time Abf1 stays bound to a specific site. The fits were done in R with the `nls` function using the formula: "`nls(ChIP ~SSasymp(time, yf, y0, log_koff), data = data)`", using all three replicates together to obtain a single fit per peak. The goodness of fit for each fit was assessed by calculating a pseudo-$R^2$, calculated as:

$$R^2 = 1 - \frac{\sum residuals^2}{\sum (y - \bar{y})^2} \qquad (2)$$

This yielded regressions with excellent $R^2$, with the lowest $R^2 = 0.65$ and the median $R^2 = 0.94$ (Fig EV2D). One of the 10 min depletion ChIP samples had much higher binding levels compared to the other 10 min samples, with a median absolute deviation more than six times as high. Upon removal of this sample all fits improved. This sample was therefore removed from all analyses. The 191 binding sites were divided into four mean residence time groups defined by the quartiles of their mean residence times (48 or 47 sites per quartile). An exponential decay model with the absolute binding levels was used rather than an exponential decay fit with log-transformed data since such a linear model cannot estimate a baseline $y_f$, yielding poor fits and inaccurate estimates for the off-rates. Dataset EV1 contains for the 191 Abf1-binding sites the coordinates of the 101 bp windows, the binding levels before and after 90 min of depletion, the estimates for $y_0$, $y_f$, $k_{off}$ and the mean residence time as well as the number of motifs, motif scores and to which residence time quartile a binding site belongs to.

## 4tU-seq expression analysis

The 4tU-seq reads were assigned to genomic features using feature-Counts from the subread package v1.6.5 (Liao *et al*, 2014). As an annotation file, transcription start site annotations from (van Bakel *et al*, 2013) were merged with the genome annotation from the Saccharomyces genome database (SGD; Cherry *et al*, 2012) containing ORFs, tRNAs, rRNAs and snRNAs. The counts from the two independent sequence runs were combined and differential expression of the genomic features (genes) was calculated using the DESeq2 package v1.10.1 (Love *et al*, 2014) in R. Only genes with a fold change of more than 1.5 with an adjusted *P*-value < 0.01 after 20 min as well as 30 min of depletion were considered differentially expressed. The fold change in mRNA synthesis was calculated relative to $t = 0$.

To model the decrease rate of mRNA synthesis, absolute transcript counts were used at each time point. They were normalized to the median number of transcripts of all samples after filtering out rRNAs. Only genes that were significantly downregulated with Abf1 binding to the promoter were used ($n = 88$). Binding peaks were assigned to genes when the summit of that peak was found within the promoter (500 bp upstream of the transcription start site). As described for the ChIP-seq data, a first-order exponential decay function (Equation 1) was used to model the changes in mRNA synthesis. In this case, $y_0$ is the expression level before depletion and $k_{off}$ is the rate with which the expression decreases to the final expression level $y_f$. For six of the genes with robust changes in mRNA synthesis, no reliable fit could be obtained and these were therefore excluded from Fig 3G ($n = 82$).

## External datasets

To assess what percentage of Abf1 binding peaks overlap with previously detected Abf1 binding sites, the peaks (peak summits $\pm$ 50 bp) detected here ($n = 948$) were compared to published datasets. Data from three different techniques were used: ORGANIC, ChEC-seq and ChIP-exo (Kasinathan *et al*, 2014; Zentner *et al*, 2015; Rossi *et al*, 2018b). For the ORGANIC data, the published bound Abf1 sites ($n = 1,068$) from the "10' MNase 80mM" samples were used (Kasinathan *et al*, 2014). For the ChEC-seq data, the Abf1 sites that were both classified as "fast" and "high scoring motif" ($n = 1,583$) were taken (Zentner *et al*, 2015). For the ChIP-exo peaks, all sites detected using the ChIP-exo protocol v5 ($n = 3,177$) were used (Rossi *et al*, 2018b).

To assess *in vivo* protection from MNase cleavage, the bigwig files "Abf1aa_V_freeMNase_ChEC" and "Abf1aa_R_freeMNase_-ChEC" were downloaded from the gene expression omnibus (GEO) dataset GSE98259 (Kubik *et al*, 2018) and smoothed with a 3 bp window. The data were centred on the Abf1 binding motif, and the average number of cleavage sites per mean residence time quartile was calculated. Subsequently, the average number of cut sites was calculated in the area of the motif ($-8$ bp to $+8$ bp from the motif midpoint) both in the presence and absence of Abf1. The cleavage ratio was calculated by taking the number of cuts in the absence of Abf1 divided by the number of cuts in the presence of Abf1. Sites without cuts in either conditions (with or without rapamycin) were excluded from the quantification (Fig 1J).

To compare synthesis rates with off-rates and binding levels, genome-wide synthesis rates were taken from (Sun *et al*, 2012), by downloading them from the researchers' website: https://www.mpibpc.mpg.de/13760807/Sc_turnover.zip.

To visualize nucleosome positioning before and after depletion of Abf1, the bigwig files "Abf1veh15" and "Abf1rapa15" were downloaded from GEO dataset GSE73337 (Kubik *et al*, 2015). The data were either centred on the Abf1 binding motif (Fig 2) or aligned on the +1 nucleosome (Figs 3B and EV3A), with +1 nucleosome positions taken from (Kubik *et al*, 2015). The average nucleosome occupancy was calculated for genes with Abf1 binding, an annotated +1 nucleosome and that were strongly downregulated (fold change > 2 at $t = 20$, $n = 44$), weakly downregulated (1.5 < fold change < 2 at $t = 20$, $n = 42$) or did not change (fold change < 1.5 at $t = 20$, $n = 112$) upon depletion of Abf1 (Fig 3B). In Fig 2, the average nucleosome occupancy was calculated per mean residence time quartile. The confidence intervals in Figs 2 and 3B are indicated by the shaded area which is calculated as the mean $\pm$ 2 × the standard error of the mean (SEM).

To visualize polymerase binding, the PAR-CLIP (photoactivatable ribonucleoside-enhanced cross-linking and immunoprecipitation) data for both the plus and minus strand from sample "Rpb2-HTB Control With Rapamycin" were downloaded from GEO dataset GSE56435 (Schaughency *et al*, 2014). As the Abf1 motif is strand-specific, the data were reoriented accordingly, meaning that when the data had to be reoriented to match the orientation of the motif (as shown in Figs EV2B and 5C), the plus and minus strand were also swapped. To calculate the roadblock efficiency, the average binding in the roadblocked peak, located at $-37$ bp $\pm$ 5 bp from the Abf1 binding motif centre, was normalized by the amount of incoming transcription (defined as the average Pol II binding in the region from $-300$ bp until $-100$ bp upstream of the Abf1 binding motif). For the quantification shown in Fig 4C, a peak was considered to be a roadblock peak when it had upstream normalized Pol II binding > 2. With this cut-off, approximately 25% of the peaks (49/191) were considered to be a roadblock for Pol II.

### Motif scoring and DNA shape analysis

The position frequency matrix from (MacIsaac *et al*, 2006) was obtained from the YeTFaSCo database (de Boer & Hughes, 2011), multiplied by a factor 1,000 and converted to a position weight matrix (PWM). The region $\pm$ 100 bp of the summit of each binding peak was searched for a match of this PWM using the matchPWM function from the Biostrings v2.38.4 package in R, using a minimum motif score of 85%. Whenever a binding peak had more than one motif match, the highest score was assigned to this binding peak. For all aggregate plots, the data were aligned to the motif with the highest motif score with all motifs in the same orientation as shown in Figs EV2B and 5C. DNA shape analysis was done on the aligned motifs $\pm$ 40 bp from the motif midpoint using DNAshapeR v1.10.0 (Chiu *et al*, 2016). The DNA shape of the bound and unbound sites from (Rossi *et al*, 2018a) was calculated as described in (Rossi *et al*, 2018a), by taking for each of the 8 motifs the top 50 and bottom 50 bound peaks and showing the average of the top 400 and bottom 400 bound peaks. For further details, see (Rossi *et al*, 2018a).

### Statistical analysis and data visualization

All statistical analyses were done using the statistical language R v3.2.2 except for the DNA shape and Venn diagram analyses which were done using R v3.5.1. The area-proportional Venn diagrams were created using the eulerr package v6.0.0 in R v3.5.1.

To visualize the binding to different genomic loci in Fig 1B, the Sushi package v1.24.0 was used (Phanstiel *et al*, 2014). All boxplots were created using R's built-in `boxplot` function, with default settings; here, the solid horizontal line represents the median, the box shows the interquartile range, and the whiskers are at the most extreme data point no further away from the closest quartile than 1.5 times the interquartile range. Differences between the four mean residence time quartiles were assessed using a one-way ANOVA followed by Tukey's honest significant difference test (Figs 1J, 4B, 5B and D). The difference between the two groups in Fig 4C was tested using a two-tailed t-test and between the groups in Fig 5A using a Wilcoxon rank-sum test, since one of the groups was deemed to deviate too much from normality.

## Data availability

The datasets and computer code produced in this study are available in the following databases:

- ChIP-Seq data: Gene Expression Omnibus GSE151692 (https://www.ncbi.nlm.nih.gov/geo/query/acc.cgi?acc = GSE151692).
- RNA-Seq data: Gene Expression Omnibus GSE151468 (https://www.ncbi.nlm.nih.gov/geo/query/acc.cgi?acc = GSE151468).
- Modeling computer scripts: GitHub (https://github.com/wdejonge/DIVORSEQ).

**Expanded View** for this article is available online.

### Acknowledgements

We thank the members of the Holstege and Kemmeren groups for their support and discussions. We thank Jeff DeMartino and Marit de Kort for technical assistance. This work was supported by the Netherlands Organisation for Scientific Research (NWO) grant 86411010 and by the European Research Council (ERC) grant 671174 DynaMech.

### Author contributions

WJJ, PK and FCPH involved in conceptualization. WJJ and MB involved in investigation. WJJ and PL involved in formal analysis. PL contributed to software. WJJ involved in data curation. PK and FCPH supervised the study and acquired funding. WJJ and FCPH contributed to writing—original draft. WJJ, MB, PL, PK and FCPH contributed to writing—review and editing.

### Conflict of interest

The authors declare that they have no conflict of interest.

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
