## [Review Process File · Molecular Systems Biology]

Genome-wide off-rates reveal how DNA binding dynamics shape transcription factor function

Wim de Jonge, Mariël Brok, Philip Lijnzaad, Patrick Kemmeren, and Frank Holstege

DOI: [10.15252/msb.20209885](https://doi.org/10.15252/msb.20209885)

Corresponding author(s): Frank Holstege (F.C.P.Holstege@prinsesmaximacentrum.nl) , Wim de Jonge (w.dejonge@prinsesmaximacentrum.nl)

Review Timeline:	Transfer from Review Commons	23rd Jul 20
	Editorial Decision:	20th Aug 20
	Revision Received:	6th Sep 20
	Accepted:	10th Sep 20

Editor: Thomas Lemberger

Transaction Report: This manuscript was transferred to The EMBO Journal following peer review at Review Commons.

**Review
COMMONS**

Response and revision

We would like to thank the editor and reviewers for their time and comments. We are happy that all three reviewers agree with the findings described in the manuscript, with all three commenting very positively on the significance and advance for the field. The suggestions of the reviewers largely involve textual changes and/or additional analyses. The majority of these suggestions have been incorporated, thereby improving the manuscript. The responses are described point-for-point in blue font below and all changes that have been made to the original manuscript are underlined. A revised manuscript (with tracked changes) accompanies this response.

Reviewer #1 (Evidence, reproducibility and clarity):

The manuscript from de Jonge et al. introduces DIVORSEQ, a method for measuring the off-rates of transcription factors (TFs). In DIVORSEQ unbound TFs are first depleted from the nucleus using the Anchors-Away technique and the decay of TF occupancy is then followed with ChIP-seq. By assuming that TFs are rapidly removed from the nucleus as they unbind the DNA, the authors can fit a first order exponential to the data and infer the off-rate (inverse of residence time). The authors focus on Abf1, an important TF in the yeast *S. cerevisiae* that regulates nucleosome free regions at promoters. Using DIVORSEQ the authors report off rates for 191 Abf1 binding sites across the yeast genome. The authors also use 4tU labeling to identify genes that lose activity following Abf1 depletion. This paper makes two claims. The first claim is that DIVORSEQ directly measures off-rates genome-wide. The second claim is that the off-rates are diverse and that binding sites with longer dwell times (slower off-rates) are more likely to be functional.

The data in the manuscript do generally support the claims. The first claim is supported by the data, but there are subtleties and caveats to the method that warrant some additional discussion. The kinetics of the Anchors-Away method sets the resolution of DIVORSEQ, so it is unclear whether 3 minutes is the shortest dwell time for Abf1 or just the limit of the resolution for DIVORSEQ. Likewise, the kinetics of formaldehyde are also slow and will impact the resolution of the assay. The paper might also benefit from a discussion of why estimates of TF dwell times in formaldehyde experiments (4.5-37 min here) are always an order of magnitude larger than estimates of dwell time in single particle tracking imaging experiments (5-40 sec)? The authors touch on this issue when contrasting bulk and single cell methods, but more text would be helpful for the reader. As a reader I wonder whether the dwell times should be interpreted on an absolute or relative scale.

The second claim is also supported by the data. The authors present strong evidence that differences in the relative dwell times at different sites are correlated with different Abf1 functions. Abf1 binding sites with longer dwell times are correlated with MNase data for nucleosome position, have larger nucleosome free regions, are more likely to be upstream of genes that show loss of activity later in the time course by TT-seq.

These are excellent points, some of which were already included in the discussion, but insufficiently. The discussion has now been expanded to address: 1) technical limitations of measuring short residence times, 2) differences in reported time scales between our method and microscopy based methods, 3) relative versus absolute values. A brief remark about using the off-rates to infer on-rates has also been added to the discussion in response to Reviewer #3.

Reviewer #1 (Significance):

Overall the manuscript makes a contribution by emphasizing the importance of quantitative kinetic analyses in systems biology and by providing a strong demonstration that such

studies are possible and informative at scale.

We appreciate that Reviewer #1 provides such a strongly positive summary of our work.

Reviewer #2 (Evidence, reproducibility and clarity):

This paper uses an ingenious and elegant system to monitor the binding stability of pioneer transcription factor Abf1 at 191 sites across the genome of *S. pombe*. The authors observe different rates of Abf1 ChIPseq signal decay after nuclear depletion of the transcription factor using a rapamycin-binding system (DIVORSEQ). The authors fit the ChIPseq decay data with a two-exponential decay model to obtain off rates and residence times, spanning from 4 to 37 minutes depending on the genomic site. The authors then rank the genomic loci in 4 groups of residence times (longest/long/short/shortest) based on quartile distribution, and which nicely coincide with degrees of MNase protection at the Abf1-bound sites. The authors then focus on finding different functional characteristics for the different observed binding stabilities: 1) sites with the shortest residence times seem to have higher nucleosome occupancy 2) there seem to be a trend for longer bound sites to elicit less variation in nascent mRNA synthesis after depletion of Abf1 3) at site with the longest residence times, Abf1 roadblock RNA pol2 4) several and good quality Abf1 motifs ensure better binding stability.

Comments:

1. A good validation for DIVORSEQ would be to show that the Rapamycin-based depletion system used here does not lead to an active stripping off of Abf1 from chromatin, which would lead to an underestimation of the binding stability measured here. Using a protein with high binding stability to chromatin as a control (eg histone H2B, histone H1, ...) FRB-GFP tagged and showing the absence of loss of GFP signal (data presented like in Figure S1D) would thus be informative.

This is a suggestion which we certainly considered at the outset. Unfortunately, histones are too highly abundant (>100,000 proteins per cell, compared to the anchor away system capable of depleting 2000 molecules/min), thus preventing a conclusive experiment. We know of no proteins similar in abundance to Abf1, with known high binding stability, similar to histones, which would be an alternative. Most importantly however, nuclear export occurs through a process called facilitated diffusion, which is not an active process (Macara, *Microbiol Mol Biol Rev* 2001; Lei & Silver, *Developmental Cell* 2002). This means that anchor-away exerts no physical force to actively strip Abf1 from DNA. Moreover, the large range of residence times determined (4.2 -33 minutes), their correspondence to factors that determine stability or that are dependent on stability, as well as the independent validation through comparison with MNase data, all provide further evidence that the method works as designed.

2. The use of the term "residence time" in the field currently applies to when individual molecules have been measured one at a time. DIVORSEQ involves measurements at an individual binding site in a large number of cells, and hence involves many molecules being measured in one data point (that being, a binding site). It would be useful to refer to their data to reflect this point, such as by considering the terms "collective residence time" or "collective off-rate" at each site.

Good point. We have changed "residence time" to "mean residence time" throughout the manuscript.

3. In Fig. S1, at early times the GFP levels of rapamycin-based depletion seem to vary

highly from cell to cell. The analysis in Fig. 1 includes a summary of the 5 min point but the primary data points in Fig. S1 do not display the 5 min point, and should. It seems that curve fitting for the variable anchor-away effect at the 5 and 10 min time points should either accommodate the cell-to-cell variation, in some fashion, or those points might best be deleted from the analysis.

We apologize for the confusion. First of all it is important to point out that the microscopy data shown in Supplemental Fig S1 is a control experiment for depletion and not as such used to derive off-rates (which was done through the ChIP-seq Abf1 depletion time course). Therefore, as indeed mentioned by the reviewer in point 2, the off-rates are collective values from many cells, measured at the same site. It is therefore not a good idea to remove the 5 and 10 min ChIP data since this will result in poorer estimates of the collective off-rates.

4. The cited correlations in Fig. 3F and G are subtle, with an r-squared of 0.5; it might be clearer if the analysis was performed in each quartile of collective off rates.

We performed the analysis as suggested. This shows that the differences between the residence time (or off-rate) quartiles are highly significant, as shown in the figures below:

This does indeed further confirm the significance of the data in Fig 3F and 3G, and shows that the off-rates are important determinants for the rate of expression changes. We prefer to show the original scatterplots in Fig 3F and 3G to be consistent with the other panels and because this more clearly shows the correspondence across all values.

5. Along these lines, the impact of Abf1 depletion on apparent nucleosome occupancy in Figs. 2A and 3B are not particularly marked. I would want to see browser views of median-change genes in each category to be convinced of significant changes.

We agree.

We have therefore included, beside the average values in Fig 2 and Fig 3B, the above figure (Supplemental Fig 3) showing example sites with different expression changes and off-rates.

6. The 191 sites focused on here are a small subset of the total Abf1 binding events. It would be best to briefly describe how those were selected in the main text, instead of having to find the details in the discussion. Along those lines, the motif analysis in fig. 5 might be good to follow Fig. 1, as an early validation that the sites are indeed binding Abf1 motifs.

Two points.

We agree and now describe the selection of these sites in the results section.

Showing the motif analysis for these sites is a bit circular because having an Abf1 motif was one of the selection criteria. Nevertheless, we included the consensus motif found for these 191 sites in Supplemental Fig 2.

7. Studies of Reb1 and Cbf1 in yeast by Poirer (eLife 2019) and of FoxA1 (Cirillo 1999 Molecular Cell) found that while on-rates to nucleosomes are lower than free DNA, off-rates are also lower. Beyond the 191 sites that the authors focus on, among the many other Abf1 sites, are some apparently coincident with a nucleosome and do those sites have a low off-rate compared to sites where Abf1 is clearly bound to nuc-free DNA?

As now explained in more detail in the results, the selection of the 191 sites was performed so as to have sites for which we could confidently obtain off-rates. Therefore, we do not feel comfortable analyzing or drawing any conclusions from other sites for which we cannot confidently measure off-rates.

8. The cells in Supp. Fig. 1C should be shown at a much higher magnification.

We agree and have increased the size of the microscopy images by 50% to make the cells more clearly visible (Supplemental Fig S1C). Unfortunately, we cannot increase the size further as this would reduce the image resolution below the 300 DPI that is required by most journals. This is obviously also related to the magnification limit of the microscope used.

9. The paper would benefit from a discussion of how their data conflict with the much shorter residence times seen from single molecule tracking experiments and perhaps note the technical limitations of the latter.

This point was also raised by Reviewer #1, we added a section to the discussion that addresses this issue.

Reviewer #2 (Significance):

The paper presents a technical advance, with a clever new method for mapping TF off-rates as ensembles, and a conceptual advance, that off-rates for TFs in chromatin may be much slower than that implied by single-molecule tracking (SMT) experiments, which lately have dominated the field.

We appreciate that Reviewer #2 recognizes the advances made in our study.

Reviewer #3 (Evidence, reproducibility and clarity):

Abf1 is a general transcription factor with DNA-binding activity, which is mostly found at promoter sites. The main goal of this work is the measurement of off-rates of Abf1-DNA binding at different Abf1 binding sites on a genome-wide level. Using *S.cerevisiae* as a model system, the authors have constructed an Abf1 anchor-away strain, in which unbound Abf1 can quickly and continuously be removed from the nucleus upon rapamycin treatment. After induction of Abf1 depletion, the authors perform an Abf1 ChIP-Seq time series experiment comprising 11 time points and 3 biological replicates. Statistical modeling of the decay of the binding signals allows the estimation of the off-rates at every sufficiently covered Abf1 binding site. The entire method is termed DIVORSEQ.

The off-rates respectively the residence time show substantial variation between binding sites, ranging between 4.5min and 37min. These rates are systematically compared to biological features of the Abf1-bound promoter regions to find factors that contribute to Abf1 stability. The authors find that

- Abf1 residence times correlate with DNA accessibility (measured by an MNase protection assay)
- Increased Abf1 binding stability is associated with larger nucleosome free regions
- Changes in mRNA synthesis rates match Abf1 binding dynamics

Since I am a bioinformatician, my review focuses on the statistics / methods aspect of the paper. The paper is well written, uses concise language and is structured in a linear fashion that does not require permanent cross-checking of statements. The Figures are not overly crowded and support the statements made in the text. They are accompanied by extensive and informative legends that make the Figures self-contained.

I appreciate the great attention the authors pay to the optimization of the experimental protocols, even small details. E.g., cell growth and the speed of growth is the major determinant of RNA expression in *S.cerevisiae*. Consequently, the application of rapamycin

in the different samples of the time course was scheduled such that at the time of harvest, the cell densities in the suspensions were approximately equal in all samples.

As to the bioinformatics analysis of the data, I have a few concerns that should be addressed (see below, major points). Nevertheless, the (almost) raw data shown in Figure 1B encourages me to believe that the estimated off-rates are qualitatively correct. Their numerical value may change a bit, though.

Major points:

1. Abf1 ChiP peak calling

Stringent filtering of the Abf1 peaks was applied to remove putatively false Abf1 binding peaks. The authors cite [Rossi et al, Genome Research 2018] to filter out peaks that do not have a G/C position at a specific motif position: "For each binding peak, we searched for motifs [...] that closely match the Abf1 consensus [...] and determined whether there was a G/C or an A/T at -8 bp from each motif."

The authors use the reversed motif sequence, compared to [Rossi et al, Genome Research 2018] - is there a particular reason for doing so? I was confused when reading [Rossi et al.]: "This result [...] indicates that the nucleotide composition at the +8 (and +9) position does not significantly affect the affinity of Abf1 for DNA, but only the ability to capture the binding event by formaldehyde cross-linking."

Most importantly, the above statement seems to imply that a lack of G/C at +8bp of the motif does not imply a false positive binding signal, it merely seems to limit the sensitivity of Abf1 binding detection at motifs that lack this position. This urgently needs clarification.

Lastly, the refined peak filtering reduces the number of considered Abf1 binding sites considerably, from 421 to 195. Therefore, I recommend showing (and analyzing) the signals from the peaks that were currently filtered out.

We apologize for the confusion regarding the motif orientation. The Abf1 motif is used in both orientations in the field. Since we used the motif from Maclsaac *et al*, 2006 to search for sites and calculate the motif scores in Fig 5B, we used that orientation for consistency, which is reversed compared to the motif used by Rossi *et al*, 2018.

We now explain the orientation chosen in the figure legend of Supplemental Fig S2.

We indeed excluded sites without a G/C at 8 bp from the Abf1 motif because formaldehyde-based methods are severely limited in detecting binding to these sites (Rossi *et al*, 2018). We found that sites without a G/C at 8bp that nevertheless yielded ChIP signal often did not show reduction of ChIP binding levels during Abf1 depletion. So, these sites are not false positives. Rather, binding at these sites (without a G/C at 8bp) likely reflects re-binding of unbound (and depleted) Abf1 to the DNA during the ChIP procedure. Therefore, estimates for off-rates at these sites cannot be trusted. This was previously already described in the discussion. We have expanded the discussion of this issue to remove any ambiguity.

2. Exponential decay fitting, modeling of k_{off} rates

It should be motivated why you include a baseline binding signal (y_f) in your exponential decay fit. If this corresponds to a baseline unspecific background signal, please explain why you do not fit one global value for y_f , but one value for each binding site and each replicate time series.

A baseline \$y_f\$ value was included in our fits because the binding levels do not go to zero, but to background levels of reads mapping to that site. We used a different \$y_f\$ for each peak because different sites have different background levels that the binding decays to (compare the last time points in Fig 1B between the different examples or the last data points in Fig 1C, 1D and 1E).

Importantly and to specifically answer the question, we did not use a separate site-specific y_f for each separate time series but used a single y_f per site across all replicates.

We now motivate the use of a site-specific \$y_f\$ in the methods section

I could not find information on how you aggregated the k_{off} estimates from the 3 replicates into one. This needs to be added.

We did not estimate k_{off} for each replicate series separately but rather fitted the model on all three replicates together and have changed the methods section to describe the fitting procedure better.

One major concern I have is the least squares procedure used to fit the data, and the R^2 value used to assess goodness of fit. According to the formula shown in the methods, all time points were fitted on the absolute scale, with equal weights. However, measurement errors in ChIP-Seq typically scale proportional to the signal intensity (apart from unspecific background signal). Consequently, the fitting method introduced here will systematically overfit early (i.e., high signal) time points. I strongly recommend log-transforming the data before performing the fit, this should approximately stabilize the variance of the observations across time points.

This is a logical suggestion which we have now carried out. However, the fits on log-transformed data do not fit the data well (see figure A below). This is a general observation because fits with log-transformed data for all 191 sites do not accurately estimate the binding levels before depletion (y_0 , figure B below). Consequently, such fits won't provide accurate off-rates.

The explanation for these poor fits is that a linear fit cannot estimate baseline y_f , so it would have to be obtained separately before doing a linear fit on the log-transformed, baseline-corrected data. Unfortunately, this strategy can only work if low binding levels can accurately be distinguished from background levels. For example, for a site with an off-rate such that the binding reduces 50% every 10 minutes, the binding levels after 20 minutes are 25%, after 30 minutes 12.5%, etc. After 90 minutes of depletion, the binding should be 0.2% of the original binding levels. ChIP does not have

the resolution to distinguish these binding levels from background. Therefore, these fits do not work well.

This is obviously a valid point and we now explain the choice of the model in the methods section.

The crucial issue however is the normalization of the individual samples. If this is not done with utmost care, it can mess up the whole analysis. The samples were rescaled to have the same coverage across the whole genome (I assume this means including all reads that were mapped to the genome). This method will only work if the vast majority of reads does NOT map to the Abf1 binding peaks: Suppose we had a perfectly specific enrichment of reads only at true Abf1 binding sites. Each ChIP sample would provide a snapshot that measures the relative abundance of the reads in a given region, compared to the total read abundance in all regions. This implies that the rescaled signal at Abf1 sites with a long residence time will artificially INCREASE with time! I strongly suggest a thorough investigation of the effects of normalization. At least, you should give the ratio of reads that map to Abf1 sites (or generally to binding peaks) vs. reads that erratically map to the genome ("unspecific Abf1 binding signals"). Assuming constant unspecific binding at all time points, this ratio might be used to perform a better normalization.

We agree that taking into account the background signal is a better normalization compared to only equalizing the total coverage. We performed the normalization as suggested, by calculating the percent of reads that comes from background regions (i.e. those not assigned to any peaks), and dividing the values that were used for the fitting by this percentage. (This essentially equalizes the background to the same levels between all samples). The off-rates calculated by this method are very strongly correlated to the originally reported off-rates ($R^2 = 0.999$, see figure below).

The actual values do change and all off-rates are slightly increased as is also visible in the figure. We therefore have remade most of the figures in the manuscript using the new values. There are no major changes because of this and this normalization method, although an improvement, does not affect any of the conclusions. As an example, the correlation between mRNA synthesis decrease rates and off-rates (Fig 3G) is shown below, before and after this normalization. The correlation is the same (with a slightly different p-value), the only difference is the position of the points on the x-axis (all are shifted to the right).

We have now applied the normalization method as suggested and all figures that show off-rates, residence times or residence time quartiles have been remade and incorporated in the manuscript. In addition, we describe this normalization to background levels in the methods section. The lower and upper residence time limits for Abf1 also changes slightly and this has been changed in the manuscript (from 4.5 – 37 minutes to 4.2 – 33 minutes)

Minor points:

3. Off-rate is defined in the equation in Figure 1A, but residence time ($1/k_{\text{off}}$) has never been defined. Just add it somewhere in the text.

We added this more clearly to the main text.

4. Fig. 1 C, D, E: According to the Methods, the 3 replicate time series have been fitted separately. You might add these three fits to the plot, in three different colors / line + point styles.

We now more clearly state in the methods that the fits were performed using all three replicate time points simultaneously.

5. Fig 1F, G: As a matter of taste, I would mirror Fig 1F and G at their main diagonal, which would give the cumulative distribution function of k_{off} respectively the residence time.

We thank the reviewer for pointing this out. This is indeed a better way to represent the data, as shown in the figures below.

We now adopted this change in the panels 1F and 1G to show the cumulative distribution.

6. Fig. 1H: Assuming steady state levels are the product of on-rate and residence time, I would rather show ChIP levels (y-axis) vs. residence time and perform a (robust) linear fit whose slope can be interpreted, up to a constant, as an average on-rate. Also, the individual

quotients ($t=0$ ChIP level divided by residence time) might, up to proportional scaling, cautiously be interpreted as the on-rate of Abf1 (probably with a huge estimation error). Still, this might be informative when comparing the set of on-rates to local DNA features. Please comment on this.

Although this is a potentially interesting avenue to explore, as the reviewer points out, such derived on-rates will likely have a huge estimation error. We are therefore reluctant to include this in the manuscript. There is a method (cross-linking kinetic analysis) that can directly measure site-specific on-rates (Poorey *et al*, 2013, Zaidi *et al*, 2017), although this has not been implemented genome-wide yet. DIVORSEQ was originally designed to be used alongside cross-linking kinetic analysis and a direct method would be a better way to determine the on-rates.

We now include this in the discussion section.

7. Fig. 2 could be made more compelling by indicating the variance of the NFR size in the four residence time groups.

We agree that the figures will benefit from an indication of the variance and have remade Fig 2 to show the confidence intervals and also adapted Fig 3A to be consistent with Fig 2.

8. Supplemental Figure S2B: The values of the residuals should scatter around 0 (on the log scale). This does not fit to the color bar (\log_2 binding) for the left and middle panels. Please correct.

We apologize for the confusion. The residuals in Fig S2B are shown as the median absolute deviation (MAD) at each time point. Since they are absolute values, they do not scatter around 0. This is described in the figure legend, but not readily visible from the plot title.

We relabeled this plot to “Median absolute deviation” to make this clear.

Reviewer #3 (Significance):

The site-specific Abf1 off-rates represent a valuable data source for future studies. The present work is among the first relating dynamic parameters of DNA binding to structural, biochemical and functional features of the underlying binding site. The DIVORSEQ method as such may be applied to other DNA binding proteins, although it is a costly method, and its adaptation will be laborious. Without any doubt, the results obtained in this study will be highly relevant for the field of transcription.

Again, a very positive statement about the significance of our findings/method for the field which we appreciate.

In addition to the changes described above, we also corrected a spelling mistake in the methods section.

The sentence was changed to:

“The 191 binding sites were divided into four residence time groups defined by the quartiles of their residence times (48 or 47 sites per quartile).”

Thank you again for submitting your work to Molecular Systems Biology. The reviewers are now fully supportive and I am pleased to inform you that we will accept your manuscript for publication pending the following minor amendments:

REFEREE REPORTS

Reviewer #1:

The authors have been diligent in addressing the reviewers' comments and in improving the analysis and discussion of the data.

Reviewer #2:

The authors have adequately addressed all points raised in my first report. In particular, they have

- implemented the background-binding based normalization procedure (which, as it turned out, only led to irrelevant changes in the estimates).
- evaluated a fitting procedure based on logarithmic binding values. They rejected it as unreliable, since it could not account for a baseline mapping rate. The fitting of an exponentially decaying signal with additive and multiplicative noise is a problem that occurs in many studies, and I still think it might be improved slightly. Yet, the estimates given in the paper are sufficiently reliable. I have no further concerns.

Corresponding Author Name: Frank C.P. Holstege

Manuscript Number: